# Maternal Underweight and Its Association with Composite Index of Anthropometric Failure among Children under Two Years of Age with Diarrhea in Bangladesh

**DOI:** 10.3390/nu14091935

**Published:** 2022-05-05

**Authors:** Rina Das, Md Farhad Kabir, Per Ashorn, Jonathon Simon, Mohammod Jobayer Chisti, Tahmeed Ahmed

**Affiliations:** 1Nutrition and Clinical Services Division, icddr,b, Dhaka 1212, Bangladesh; rina.das@icddrb.org (R.D.); md.farhad.kabir@gmail.com (M.F.K.); tahmeed@icddrb.org (T.A.); 2World Health Organization, Avenue Appia 20, 1211 Geneva, Switzerland; per.ashorn@tuni.fi (P.A.); jonleesimon@gmail.com (J.S.); 3Faculty of Medicine and Health Technology, Tampere University, 33240 Tampere, Finland

**Keywords:** CIAF, maternal undernutrition, ABCD trial, maternal BMI

## Abstract

Malnutrition in women has been a long-standing public health concern, with serious effects on child survival and development. Maternal body mass index (BMI) is an important maternal nutritional indicator. There are few published studies although child anthropometric failures do not occur in isolation and identifying children with single versus several co-occurring failures can better capture cases of growth failure in combination: stunting, wasting, and underweight. In the context of multiple anthropometric failures, traditional markers used to assess children’s nutritional status tend to underestimate overall undernutrition. Using the composite index of anthropometric failure (CIAF), we aimed to assess the association between maternal undernutrition and child undernutrition among children with diarrhea under the age of two and to investigate the correlates. Using 1431 mother-child dyads from the Antibiotic for Children with Diarrhea (ABCD) trial, we extracted children’s data at enrollment and on day 90 and day 180 follow-ups. ABCD was a randomized, multi-country, multi-site, double-blind, placebo-controlled clinical trial. The Bangladesh site collected data from July 2017 to July 2019. The outcome variable, CIAF, allows combinations of height-for-age, height-for-weight, and weight-for-age to determine the overall prevalence of undernutrition. The generalized estimating equation was used to explore the correlates of CIAF. After adjusting all the potential covariates, maternal undernutrition status was found to be strongly associated with child undernutrition using the CIAF [aOR: 1.4 (95% CI: 1.0, 1.9), *p*-value = 0.043] among the children with diarrhea under 2 years old. Maternal higher education had a protective effect on CIAF [aOR: 0.7 (95% CI: 0.5, 0.9), *p*-value = 0.033]. Our study findings highlight the importance of an integrated approach focusing on maternal nutrition and maternal education could affect a reduction in child undernutrition based on CIAF.

## 1. Introduction

In low- and middle-income settings, maternal and under-5 child malnutrition is common, resulting in significant increases in under-5 mortality and the morbidity burden. Despite the success of numerous nutritional initiatives at the national and international levels, maternal malnutrition remains a serious public health concern with negative effects on children’s health. Previous studies have shown that maternal malnutrition is commonly associated with childhood malnutrition [1,2,3]. Child undernutrition encompasses stunting, wasting, and being underweight as one form of the condition known as malnutrition. Stunting and/or wasting may occur in underweight children, and some children may have all three forms of anthropometric failure at the same time [4]. As a result, none of these traditional nutritional indicators can accurately predict the entire burden of undernutrition and the combined estimate of undernutrition among children under the age of two. Any low and middle-income country, like Bangladesh, needs a joint estimate of anthropometric failure to comprehend the true burden of undernutrition.

For the past decade, several studies have investigated the impact of diarrheal illnesses on malnutrition [5,6,7]. They discovered that malnourished children had an increased risk of diarrhea; this is a new demonstration of a causal role for diarrheal infections in malnutrition that previous studies had not been able to establish, and these studies only included children after the critical first two years of life, when malnutrition begins [8].

Undernutrition in the first five years of life causes morbidity in 200 million children, including cognitive and physical impairment [9,10]. Many children in low- and middle-income countries (LMICs) suffer from different anthropometric failures simultaneously. [11]. Anthropometric failures do not occur in isolation [12,13]; discriminating between children with single versus several co-occurring failures can help identify cases of duel or multiple nutritional deficiencies more precisely which are often overlooked in nutritional surveys. Children who are classified as stunted by conventional definition may also, include those who have single (stunting only) and multiple failures (e.g., stunted and underweight, or even all three failures at the same time). On the other hand, children who are classified as not stunted by conventional definition may still include those who have other types of failure. Different combinations of single and multiple failures are likely due to different etiologies, necessitating different treatment approaches [14,15,16]. Furthermore, the composite indicator of anthropometric failure (CIAF) model (presence of any anthropometric form of undernutrition) has been utilized in various research studies [12], with the conclusion that CIAF consistently detects a higher percentage of undernourished children than other indices individually [10,13]. Furthermore, the CIAF allows for more precise identification of the population’s nutritionally sensitive segment. By recognizing duplicate or many failures, CIAF disaggregation has the potential to improve the efficacy of a nutritional intervention program.

According to the Bangladesh Demographic and Health Survey (BDHS) 2011, stunting, wasting, and being underweight are still widespread among Bangladeshi children under the age of five. The estimates of undernutrition provided by these traditional indicators may overlap, resulting in a lack of a comprehensive estimate of undernutrition for any country [13]. However, according to a systematic review conducted in LMIC countries, many children have numerous anthropometric failures, putting them at a higher risk of morbidity and mortality. Three anthropometric fails in a combination put a child at a 12-fold higher risk of death [17].

A joint estimate of anthropometric failure is crucial to understanding the real burden of undernutrition for any LMIC country like Bangladesh. The estimate of CIAF helps to modify the existing intervention or develop a new nutritional program targeting specific populations. Prior studies on the factors that influence a child’s nutritional outcomes have mostly focused on immediate issues such as nursing, medical treatment [18], and birth spacing [19]. Furthermore, socioeconomic variables like household economic status, maternal and paternal education, mothers’ health-seeking behavior, sanitation, fertility, and maternal stature have a role in the gap in stunting prevalence in Bangladesh [20]. However, little attention has been paid to intergenerational factors that may predispose a child to increased health risks. BMI can be a useful marker for finding intergenerational health relationships since it reflects a mother’s health stock accumulated over her life path, particularly social and environmental exposures in her early infancy [21]. Although several studies have found links between maternal BMI and child stunting [22,23], more research needs to be undertaken to find out the association between maternal underweight and child duel or multiple anthropometric failures.

To fulfill this gap in the literature, first of all, we classify child nutritional indicators into seven groups: no failure; wasting only; wasting and underweight; wasting, stunting, and underweight; stunting and underweight; stunting only; underweight only [12]. Then we categorized them into three groups based on their CIAF scores: single, double, and triple failures, and we aimed to evaluate the association of maternal undernutrition with child undernutrition using CIAF among children aged 2 to 23 months in Bangladesh.

## 2. Materials and Methods

### 2.1. Study Site

Related data were extracted from the Antibiotic for Children with Diarrhea (ABCD) trial, Bangladesh site database. The location of the ABCD Bangladesh site was in icddr,b Dhaka Hospital, and Mirpur Treatment Centre, situated in Dhaka, capital of Bangladesh. The Dhaka Hospital of icddr,b is the largest diarrheal disease hospital in the world. An average of 250 patients are treated in the hospital each day. Details about the study site have been reported elsewhere [24]. The Mirpur Treatment Center of icddr,b is a 50-bed hospital treating diarrheal patients for the last 10 years.

### 2.2. Study Design and Study Participants

The design and methodology of the ABCD trial were mentioned earlier [25]. Briefly, data were extracted from children enrolled at the ABCD Bangladesh site between 1 July 2017, and 10 July 2019. ABCD trial was a randomized, multi-country, multi-site, double-blind, placebo-controlled clinical trial which was done in 7 countries including Bangladesh among 2–23 months old children who were hospitalized with moderate to severe dehydration [26]. For recruitment, the study used a well-defined standardized approach [25]. Following the identification of a possibly eligible child, the study personnel conducted an eligibility screening [25], using a standardized screening form. If anthropometric criteria were met, the child was enrolled if he or she had diarrhea but no evidence of dehydration. They were kept under monitoring if the child showed signs of “some” or “severe” dehydration, or if the child required immediate medical attention. Oral and/or intravenous rehydration was provided during this “stabilization” period, and all acute problems were treated using standard treatment according to the WHO Pocket Book of Hospital Care for Children, 2013 [27]. If the child was eligible, he or she was enrolled after rehydration and urgent care were completed [25]. Children enrolled in the trial were followed up for 90 days and 180 days post-enrolment. Participants were followed up in the hospital on days 90 and 180 to determine their vital status, hospitalizations, and health, as well as their dietary status (weight, length, and MUAC). Maternal weight and height were recorded at initial enrolment.

### 2.3. Variable of Interest

Based on a comprehensive literature review, previous descriptive studies, and the availability of data in our investigation, several variables were considered explanatory variables (Figure 1). The outcome variables for this study were indicators of child nutritional status and CIAF (presence of any anthropometric form of undernutrition).

#### 2.3.1. Anthropometric Measurements

These were carried out following the methods specified in the WHO module on measuring a child’s growth [25] at enrollment, 90 days, and 180 days post-enrollment. Anthropometry measurements were taken after rehydration and stabilization as required. Weight was measured by two independent observers using an Electronic Baby Weighing Scale M118600 with a sensitivity of 10 g; the length was measured using a Baby length measuring board MZ10040 to the nearest 0.1 cm. Calibration of the instruments to measure length and weight was done daily. The average was calculated based on three readings. The Z-score scale = (observed value − average value of the reference population)/standard deviation value of the reference population was used to calculate length-for-age (LAZ), weight-for-age (WAZ), and weight-for-length (WLZ) z scores as indicators of child nutritional status according to 2006 WHO Standards for Children [28]. For the ABCD trial, we enrolled severely stunted or moderately wasted, or dehydrated children. Severely wasted children were excluded and referred for immediate nutritional support and antibiotic therapy.

#### 2.3.2. The Composite Indicator of Anthropometric Failure (CIAF)

The CIAF is an aggregated single anthropometric measure that provides an overall estimate of undernourished children [13]. CIAF has been proposed by Svedberg in 2000 [12] and modified by Nandy in 2005 [13]. An additional sub-group (named Y) has been added to the previous model’s six sub-groups of anthropometric failure (designated A–F) [13]. In the ABCD Bangladesh site data set, initially, we calculated stunting, wasting, and underweight from WAZ, WHZ, and LAZ where stunting (F) was defined as a LAZ  <  −2, wasting (B) as a WLZ  <  −2, and underweight (Y) as a WAZ  <  −2. Then we further generated another four variables with different combinations of different anthropometric failures as children having: wasting and underweight (C); stunting and underweight (E); wasting, stunting and underweight (D), and those who were adequately nourished (A = no anthropometric failure). After that, if a child had experienced any of the anthropometric failures, he or she was considered undernourished as measured by the CIAF. CIAF was measured as a binary response as being nourished/ no anthropometric failure (coded as 0) and undernourished (coded as 1; if the child had at least one form of undernutrition) (Table 1).

#### 2.3.3. Maternal Underweight

Maternal weight was measured using an electronic weight machine (Electronic Floor Scale M320600) with a sensitivity of ±10 g; the length was measured using a measuring tape and board (Mechanical Height MZ10017) to the nearest 0.1 cm. Maternal weight and height were measured and expressed as body mass index [BMI – weight (kg)/height (m^2^)]. Maternal underweight was defined as a BMI < 18.5 kg/m^2^ [29].

#### 2.3.4. Maternal Factors

The variables relevant to maternal factors included current age, maternal BMI, maternal education (illiterate, below primary, and primary and above), number of deliveries, and decision-making power regarding health care seeking (by herself and others).

#### 2.3.5. Age-Appropriate Breastfeeding

Categorized into three groups, exclusively breastfed (exclusively breastfed: infants should receive only breast milk except for prescribed medications, vitamins, and minerals) [30], partially breastfed: both breastfeeding and complementary feeding), and non-breastfed.

#### 2.3.6. Household Visit

ABCD field staff members visited each enrolled child’s household on discharge from the hospital. During these household visits, detailed household data were obtained. The household demographic variables included type of floor (cemented or non-cemented), type of toilet (sewer and non-sewer), and source of drinking water (piped water and others).

#### 2.3.7. Collection of Stool Samples and Stool Microbiology

Stool samples for the ABCD trial were evaluated using the ABCD laboratory process methodology for each child at the time of enrolment [25]. Diarrheal etiology identification was facilitated by quantitative molecular diagnostic methods.

### 2.4. Ethical Consideration

The WHO Ethics Review Committee and the icddr,b Ethics Committee both authorized the trial. All of the methods were carried out in accordance with the relevant norms and regulations. Before enrolling, the research staff got written informed consent from parents/caregivers.

### 2.5. Statistical Analysis

The data were summarized using descriptive statistics such as mean/standard deviations for quantitative factors and frequencies/proportions for qualitative variables, and it was separated by maternal nutritional status and multiple anthropometric failures. CIAF was dummy coded such that the CIAF was compared to the reference category of ‘no failure’. A generalized estimating equation (GEE) using the binomial family and logit link function was used to analyze the relationship between maternal underweight status and indicators of all seven-child nutritional status. Exchangeable correlation with default standard errors was utilized as the within-group correlation structure for the population average panel data model, and the repeated measure was applied as the time variable using the xtset function in STATA. All covariates were used in the multiple GEE model to achieve an adjusted final model (child age, sex, breastfeeding status, number of under 5 children in the house, maternal age, maternal education, type of floor, source of drinking water, toilet facility, pathogen (Campylobacter: >48 percent present) isolated from stool). The variance inflation factor (VIF) was calculated to detect multicollinearity, and no variable with a VIF value greater than 5 was discovered in the final model. The criterion of significance was set at 0.05, and 95% confidence intervals were calculated to determine the directions and strength of the effects. All data were analyzed using version 15.0 IC of STATA (College Station, TX, USA: Stata Corp LLC).

## 3. Results

A total of 1431 mother-child dyads were enrolled in this study. Overall, 263 mothers were underweight and had a BMI of ≤18.5. Among the children of underweight mothers, 46.8% were stunted, 57% were wasted, and 76% were underweight (Figure 2).

86.3% of all underweight mothers were from the 18–29 y age group, 66.4% of mothers had more than primary level education, 12.6% of underweight mothers had more than two parity, 81.8% of underweight mothers can take healthcare-seeking decisions by themselves. About 95% of underweight mothers lived in a house where the floor was made of cement. The descriptive findings for this dataset segregated by maternal nutritional status are shown in Table 2.

Table 3 presents descriptive statistics of the 1431 dyads from the Bangladesh site. Around 60% of children were from the 2–11 months age group and 57.5% were males. Only 94.1% of children were breastfed. About 18.38% of mothers were underweight, most of the mothers (76.1%) were from the 18–29 years age group, and 67.5% of the mothers were educated more than the primary level. The majority of the households used piped water (90.1%), had a sewer toilet system (77.1%), and had cemented floor (95.5%) in the house.

In the generalized estimating equation, the odds of CIAF [aOR: 1.4, (95% CI: 1.0, 1.9); *p*-value: 0.043] were less likely among younger children [aOR: 0.97, (95% CI: 0.96, 0.99); *p-* value: <0.001] and more likely in male children [aOR: 1.8, (95% CI: 1.4, 2.2); *p*-value: <0.001]. The children whose mother’s education [aOR: 0.7, (95% CI 0.5, 0.97); *p*-value: 0.033] level was primary and above were protective of CIAF (Table 4).

After adjusting for the covariates (child age, sex, breastfeeding status, number of under-5 children at house, maternal age, maternal education, type of floor, source of drinking water, toilet facility, pathogen (Campylobacter) in >48% of cases isolated from stool) in the generalized estimating equation (GEE) maternal underweight was associated with child stunting, wasting and underweight [aOR: 1.5 (95% CI: 1.0, 2.0); *p*-value = 0.032] and CIAF [aOR: 1.4 (95% CI: 1.0, 1.9; *p*-value = 0.043)] among the children (Table 5).

## 4. Discussion

This study looked at data from the ABCD trial to see if there was a link between maternal underweight and child nutritional status using the CIAF scale, as well as identify variables. After accounting for the relevant factors in our data set, maternal underweight (BMI 18.5) was found to be associated with the triple failure of stunting, wasting, and underweight status, as well as CIAF in children under the age of two years who presented to health facilities with diarrhea. Underweight mothers were more likely to have underweight children [31] and food insecurity, caloric deficits, and poverty all influence households; the link between maternal underweight and poor child growth is well known [9,32]. More importantly, using underweight as an aggregate indicator underestimates the severity of undernutrition because it is the result of stunting and wasting, not the sum of the two. Many children have several anthropometric failures, according to a systematic review undertaken in poor countries, which leads to a higher risk of morbidity and mortality [17]. When compared to a control group, studies in Asian countries have discovered a link between stunting and wasting. According to a review of research, underweight children will experience stunting and/or wasting, and some children may experience all three forms of anthropometric failure at the same time [4]. As a result, none of these traditional nutritional indicators can accurately quantify the entire burden of undernutrition and the combined estimate of undernutrition among children under the age of five. CIAF not only gives a single measure of a population’s undernutrition burden but also aids in the detection of children with numerous anthropometric inadequacies, allowing for targeted interventions. Apart from that, CIAF has significant consequences for policymakers and organizations attempting to reach international nutrition targets, as they lack a platform on which to build and assess nutrition programs, as well as determine the scope of their coverage [33].

Our study has applied the CIAF scale for estimating the overall burden of child undernutrition and identifying some covariates. This prevalence of CIAF was higher in many developing countries including India [34,35], Ethiopia [36], and Nepal [37], and lower in Tanzania, Zimbabwe, Bolivia, and Peru [13] than the estimates in our study. The high rate of child undernutrition may impact the higher burden of morbidity, which results in higher rates of mortality among the affected children [38]. Furthermore, the CIAF provides a single measure of community undernutrition and aids in the detection of children with various anthropometric inadequacies. Given that it distinguishes between overall and complete anthropometric failure, CIAF may be a better indicator of nutritional status than standard measurements of stunting, underweight, and wasting, independently. We found gender differences for CIAF; male children were more prone to develop undernutrition. Being male gender was identified as a risk factor for malnutrition in several studies [39,40,41]. Males are more stunted than females, according to a meta-analysis conducted in Sub-Saharan Africa, implying that males are more prone to health inequities than females [42]. Due to the great value put on women’s agricultural labor in Sub-Saharan Africa, Svedberg claimed that the minor anthropometric advantage demonstrated by girls, women, or both in many nations may reflect a historical pattern of preferential treatment of females [43]. Child malnutrition rates in India, Pakistan, and Bangladesh are greater than in Sub-Saharan Africa. This is directly related to gender discrimination in South Asia [44]. In Afghanistan, Bangladesh, India, and Pakistan, boys were 16 percent to 36 percent more likely than girls to be wasted [45]. This contradicts some early research on child malnutrition in South Asia, which claimed that girls were more likely than boys to be stunted or wasted [46,47]. However, since the mid-1990s, all Demographic and Health Surveys in Bangladesh, India, and Nepal have shown that the prevalence of wasting is consistently higher among boys than girls. This is now also seen to be true on a worldwide scale [48]. This is a relationship that needs to be investigated further. Furthermore, according to epidemiological studies in neonatology and cohorts of pre-term infants and children, both morbidity and mortality are consistently higher in males than females in early life, with the differences persisting after adjusting for gestational age and body size, and being more pronounced in pre-term subjects [49,50]. The reported male majority in both symptomatic and asymptomatic morbidity, however, shows that boys are more vulnerable in general [51]. According to another study conducted in Bangladesh, socioeconomic gaps in stunting have widened over time [52]. However, a study on BDHS data showed there was no gender difference for undernutrition [53] among Bangladeshi children. A study conducted in Bangladesh found the male preponderance in rotavirus diarrhea patients attending the hospital [54]. There was a slight male predominance observed among the diarrheal patients having pneumonia in icddr,b Dhaka Hospital [55]. This might be the cause of male predominance in our study.

In our study, we found that CIAF was more prevalent among older-age children which is consistent with other previous studies. Studies from Ethiopia [56] and Burkina Faso [57] show that the number of children who are undernourished rises with their age. Food insecurity and malnutrition are clearly linked, according to a community-based study. Children between the ages of 6 and 59 months are at a higher risk of malnutrition [58]. This might be the same cause behind stunting which was found more in the older age group of children.

Prior studies have found that the relationship between maternal education and child nutritional status varies depending on socioeconomic position [59]. However, in our study, we found maternal primary and above primary level education has a protective effect on the CIAF. In a Bolivian study, they claimed that the most important links between maternal education and child nutritional status are socio-economic factors [60].

The study’s strengths include large sample size and strong methodology, which comprises proper sampling approaches, adequate training, exact anthropometry devices, thorough quality control, and a very low loss to follow-up (less than 1%). Children who presented with diarrhea to first-level health facilities in Bangladesh and LMICs were included in the study. Our study has its limitations. ABCD study children needed to have either severe stunting, moderate wasting, or some/ severe dehydration. It is possible that any association observed in the current analysis may not apply to children who have diarrhea but no additional mortality risk that includes severe acute malnutrition. The unavailability of detailed socio-economic status data is another limitation of the study, so we have used some of the household data as a proxy for determining the socio-economic condition of the enrolled children.

## 5. Conclusions

Based on CIAF, the findings of this study provide an overall burden of maternal undernutrition. Child undernutrition depends on the lower education status of the mother; beyond infantile age and male children are more prone to develop anthropometric failure. To lessen the burden of child malnutrition and meet Bangladesh’s sustainable development goal of improved nutrition by 2030, proper intervention programming with a focus on maternal undernutrition may be critical.

## Figures and Tables

**Figure 1 nutrients-14-01935-f001:**
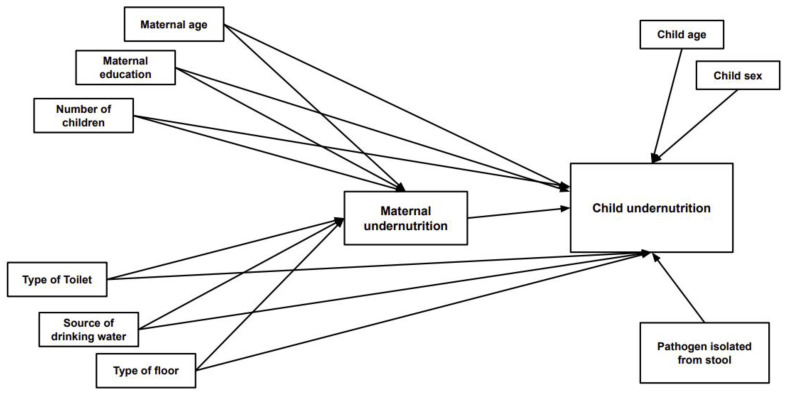
Factors associated with maternal and child undernutrition.

**Figure 2 nutrients-14-01935-f002:**
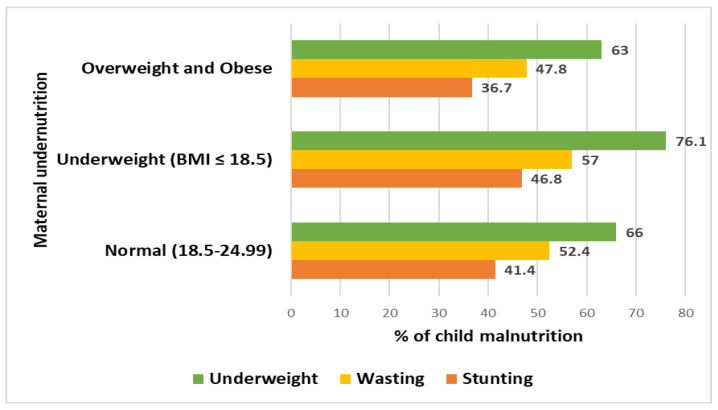
Distribution of child malnutrition based on maternal BMI.

**Table 1 nutrients-14-01935-t001:** Classification of a composite index of anthropometric failure to assess undernutrition among the children under 2 years [13].

Group	Descriptions (Child Undernutrition)	Wasting	Stunting	Underweight
A	No failure	No	No	No
B	Only wasting	Yes	No	No
C	Wasting and underweight	Yes	No	Yes
D	Stunting, wasting, underweight	Yes	Yes	Yes
E	Stunting and underweight	No	Yes	Yes
F	Only stunting	No	Yes	No
Y	Only underweight	No	No	Yes

**Table 2 nutrients-14-01935-t002:** General characteristics of the children by maternal BMI.

Indicators	Normal (BMI: 18.5–24.99) *n* (%)	Underweight (BMI ≤ 18.5) *n* (%)	Overweight and Obese (BMI ≥ 25) *n* (%)	Overall *n* (%)
**Child characteristics**				
**Sex (male)**	463 (57.9)	165 (62.7)	194 (52.7)	822 (57.4)
**Age in months**				
2–11	495 (61.9)	145 (55.1)	228 (62.0)	868 (60.7)
12–24	305 (38.1)	118 (44.9)	140 (38.0)	563 (39.3)
**Breastfeeding**				
Exclusive	11 (1.4)	2 (0.8)	6 (1.6)	19 (1.3)
Partial	746 (93.3)	251 (95.4)	330 (89.7)	1327 (92.7)
non-breastfed	43 (5.4)	10 (3.8)	32 (8.7)	85 (5.9)
**Stunting**	331 (41.4)	123 (46.8)	135 (36.7)	589 (41.2)
**Wasting**	411 (51.4)	150 (57.0)	176 (47.8)	737 (51.5)
**Underweight**	528 (66.0)	200 (76.1)	232 (63.0)	960 (67.1)
**Maternal characteristics**				
**Maternal Age in years**				
<18 y	20 (2.5)	9 (3.4)	3 (0.8)	32 (2.2)
18–29 y	612 (76.5)	227 (86.3)	250 (67.9)	1089 (76.1)
>30	168 (21.0)	27 (10.3)	115 (31.3)	310 (21.7)
**Maternal education**				
Illiterate (0)	155 (19.5)	51 (19.5)	64 (17.5)	270 (19.0)
Below primary (<5)	122 (15.3)	37 (14.1)	35 (9.6)	194 (13.6)
Primary and above (≥5)	520 (65.2)	174 (66.4)	267 (73.0)	961 (67.4)
**Parity**				
Two	635 (79.6)	229 (87.4)	256 (69.8)	1120 (78.5)
>Two	163 (20.4)	33 (12.6)	111 (30.3)	307 (21.5)
**Number of under 5 children in household**				
One	662 (82.8)	220 (83.6)	313 (85.1)	1195 (83.5)
Two	129 (16.1)	41 (15.6)	51 (13.7)	221 (15.4)
Three and more	9 (1.1)	2 (0.8)	4 (1.1)	15 (1.1)
**Health care-seeking decisions taken**				
By herself	668 (83.5)	215 (81.8)	287 (78.0)	1170 (81.8)
others	132 (16.5)	48 (18.3)	81 (22.0)	261 (18.2)
**Household characteristics**				
**Improved toilet facility**				
Sewer	614 (77.0)	207 (78.7)	280 (76.1)	1101 (77.1)
non-sewer	183 (23.0)	56 (21.3)	88 (23.9)	327 (22.9)
**Drinking water**				
Piped	717 (89.6)	236 (89.7)	336 (91.3)	1289 (90.1)
Others	83 (10.4)	27 (10.3)	32 (8.7)	142 (9.9)
**Type of floor**				
Cemented	754 (94.3)	251 (95.4)	362 (98.4)	1367 (95.5)
Non-cemented	46 (5.8)	12 (4.6)	6 (1.6)	64 (4.5)

Stunting was defined as a LAZ  <  −2, wasting as a WLZ  <  −2, and underweight as a WAZ  <  −2; BMI: Body mass index.

**Table 3 nutrients-14-01935-t003:** Distribution of maternal, child, and household characteristics by the 7 anthropometric failures at baseline.

Indicators	Overall *n* (%)	No Failure (A) *n* = 307 (%)	Stunted Only (F) *n* = 54 (%)	Underweight (Y) *n* = 55 (%)	Wasted Only (B) *n* = 110 (%)	Stunted and Underweight (E) *n* = 278 (%)	Wasted and Underweight (C) *n* = 370 (%)	Stunted, Wasted, and Underweight (D) *n* = 257 (%)
**Child characteristics**								
**Sex (male)**	822 (57.4)	127 (41.4)	41 (75.9)	24 (43.6)	64 (58.2)	177 (63.7)	212 (57.3)	177 (68.9)
**Age in months**								
2–11	868 (60.7)	211 (68.7)	40 (74.1)	45 (81.8)	77 (70)	184 (66.2)	211 (57.0)	100 (38.9)
12–24	563 (39.3)	96 (31.3)	14 (25.9)	10 (18.2)	33 (30)	94 (33.8)	159 (43)	157 (61.1)
**Breastfeeding**								
Breastfed	1346 (94.1)	285 (92.8)	50 (92.6)	52 (94.6)	105 (95.5)	258 (92.8)	354 (95.7)	242 (94.2)
Non-breastfed	85 (5.9)	22 (7.2)	4 (7.4)	3 (5.5)	5 (4.6)	20 (7.2)	16 (4.3)	15 (5.8)
**Maternal characteristics**								
**Maternal Age in years**								
<18 y	32 (2.2)	4 (1.3)	3 (5.6)	0	1 (0.9)	7 (2.5)	8 (2.2)	9 (3.5)
18–29 y	1089 (76.1)	234 (76.2)	46 (85.2)	13 (78.2)	74 (67.3)	212 (76.3)	286 (77.3)	194 (75.5)
>30	310 (21.7)	69 (22.5)	5 (9.3)	12 (21.8)	35 (31.8)	59 (21.2)	76 (20.5)	54 (21.0)
**Maternal underweight**								
Yes	263 (18.4)	40 (13.0)	7 (13)	11 (20)	16 (14.6)	55 (19.8)	73 (19.7)	61 (23.7)
No	1168 (81.6)	267 (87)	47 (87.0)	44 (80)	94 (85.5)	223 (80.2)	297 (80.3)	196 (76.3)
**Maternal education**								
Illiterate (0)	270 (19)	57 (18.6)	18 (34)	8 (14.6)	13 (11.8)	63 (22.7)	59 (16)	52 (20.5)
Below primary (<5)	194 (13.6)	41 (13.4)	12 (22.6)	10 (18.2)	74 (12.7)	45 (16.2)	42 (11.4)	30 (11.8)
Primary and above (≥5)	961 (67.4)	208 (68)	23 (43.4)	37 (67.3)	83 (75.5)	169 (61.0)	269 (72.7)	172 (67.7)
**Parity**								
Two	1120 (78.5)	239 (78.1)	42 (79.3)	41 (74.6)	86 (78.2)	212 (76.3)	301 (81.4)	199 (78.0)
>Two	307 (21.5)	67 (21.9)	11 (20.8)	14 (25.5)	24 (21.8)	66 (23.7)	69 (18.7)	56 (22)
**Number of under 5 children in household**								
One	1195 (83.5)	251 (81.8)	43 (79.6)	45 (81.8)	95 (86.4)	223 (80.2)	324 (87.6)	214 (83.3)
Two	221 (15.4)	52 (17)	10 (18.5)	10 (18.2)	15 (13.6)	50 (18)	43 (11.6)	41 (16)
Three and more	15 (1.1)	4 (1.3)	1 (1.9)	0	0	5 (1.8)	3 (0.8)	2 (0.8)
**Health care-seeking decisions taken**								
By herself	1170 (81.8)	247 (80.5)	12 (22.2)	47 (85.5)	92 (83.6)	217 (78.1)	315 (85.1)	210 (81.7)
others	261 (18.2)	60 (19.5)	42 (77.8)	8 (14.6)	18 (16.4)	61 (21.9)	55 (14.9)	47 (18.3)
**Household characteristics**								
**Improved toilet facility**								
Sewer	1101 (77.1)	237 (77.5)	8 (14.8)	43 (79.6)	84 (76.4)	206 (74.1)	294 (79.5)	191 (74.6)
non-sewer	327 (22.9)	69 (22.6)	46 (85.2)	11 (20.4)	26 (23.6)	72 (25.9)	76 (20.5)	65 (25.4)
**Drinking water**								
Piped	1289 (90.1)	282 (91.9)	10 (18.5)	48 (87.3)	102 (92.7)	249 (89.6)	331 (89.5)	233 (90.7)
Others	142 (9.9)	25 (8.1)	44 (48.6)	7 (12.7)	8 (7.3)	29 (10.4)	39 (10.5)	24 (9.3)
**Type of floor**								
Cemented	1367 (95.5)	295 (96.1)	3 (5.6)	2 (3.6)	107 (97.3)	256 (92.1)	358 (96.8)	247 (96.1)
Non-cemented	64 (4.5)	12 (3.9)	51 (94.4)	53 (96.4)	3 (2.7)	22 (7.9)	12 (3.2)	10 (3.9)

**Table 4 nutrients-14-01935-t004:** Factors associated with the CIAF (no failure as the reference group).

Indicators	Crude OR (95% CI)	*p*-Value	aOR (95% CI)	*p*-Value
**Maternal BMI**				
Normal and overweight	Ref		Ref	
Underweight	1.5 (1.2, 1.9)	0.002	1.4 (1.0, 1.9)	0.043
**Maternal age**	0.9 (0.96, 0.99)	0.021	0.9 (0.9, 1.0)	0.572
**Maternal education**				
Illiterate	Ref		Ref	
Below primary	0.9 (0.6, 1.3)	0.508	0.9 (0.7, 1.5)	0.949
Primary and above	0.7 (0.5, 0.9)	<0.001	0.7 (0.5, 0.9)	0.033
**The pathogen identified in stool**				
**Campylobacter**				
No	Ref		Ref	
Yes	1.2 (0.9, 1.5)	0.118	1.2 (0.9, 1.5)	0.167
**Breastfeeding status**				
Breastfed	Ref		Ref	
Non-breastfed	0.7 (0.5, 1.0)	0.060	0.6 (0.2, 1.5)	0.250
**Child age** (months)	0.9 (0.96, 0.98)	<0.001	0.9(0.96, 0.99)	<0.001
**Child sex**				
Female	Ref		Ref	
Male	1.7 (1.4, 2.1)	<0.001	1.8 (1.4, 2.2)	<0.001
**Under 5 children at house**	1.1 (0.9, 1.3)	0.579	1.0 (0.8, 1.4)	0.896
**Toilet facility**				
Non-sewer	Ref		Ref	
Sewer	0.9 (0.8, 1.2)	0.555	0.9 (0.7, 1.2)	0.449
**Source of drinking water**				
Non-piped	Ref		Ref	
Piped	0.8 (0.6, 1.1)	0.183	0.8 (0.5, 1.3)	0.369
**Floor**				
Non-cemented	Ref		Ref	
Cemented	0.8 (0.5, 1.3)	0.402	0.9 (0.5, 1.6)	0.633

CIAF: a composite indicator of anthropometric failure; 95% CI: 95% confidence interval; aOR: adjusted Odds Ratio; outcome variable: CIAF.

**Table 5 nutrients-14-01935-t005:** Association between different anthropometric failure categories and maternal undernutrition.

Categories of Anthropometric Failure	* aOR (95% CI)	*p*-Value
	Maternal Undernutrition	
Wasted only	0.9 (0.5, 1.7)	0.827
Wasted and underweight	1.1 (0.8, 1.7)	0.356
Stunted, wasted and underweight	1.5 (1.0, 2.0)	0.032
Stunted and underweight	1.1 (0.8, 1.4)	0.647
Stunted only	0.8 (0.5, 1.2)	0.287
Underweight only	0.9 (0.6, 1.7)	0.955
CIAF	1.4 (1.0, 1.9)	0.043

***** adjusted for child age, sex, breastfeeding status, number of under 5 children at house, maternal age, maternal education, type of floor, source of drinking water, toilet facility, pathogen (Campylobacter) isolated from stool; 95% CI: 95% confidence interval; aOR: adjusted Odds Ratio; CIAF: a composite indicator of anthropometric failure; Stunting was defined as a LAZ < −2, wasting as a WLZ  <  −2 and underweight as a WAZ  <  −2.

## Data Availability

Request for data from deay@who.int.

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
