# Peer review of "Maternal Underweight and Its Association with Composite Index of Anthropometric Failure among Children under Two Years of Age with Diarrhea in Bangladesh"

_nutrients, 2022, doi:10.3390/nu14091935_

Round 1

Reviewer 1 Report

In this study, Das and colleagues have demonstrated the correlation between undernutrition in children and maternal undernutrition status. While I don’t see any major concerns with this study and authors have to be commended for this informative study. Overall, the manuscript is well written. I have the following minor questions.

  1. The article has some unconventional language style – for example ‘women’s malnutrition’ can be referred to as ‘malnutrition in women’; similarly, ‘diarrheal children’ – ‘children with diarrhea’
  2. Figure 1 – why maternal job, family income, maternal/second hand smoking, micronutrient supplementation (both for mother and child), and birth weight, are not taken in to consideration.
  3. For the children in 6-24 months category, mid-arm circumference would have been a good indicator. Why this was not included in the ABCD trial?

Author Response

In this study, Das and colleagues have demonstrated the correlation between undernutrition in children and maternal undernutrition status. While I don’t see any major concerns with this study and authors have to be commended for this informative study. Overall, the manuscript is well written. I have the following minor questions.

1. The article has some unconventional language style – for example ‘women’s malnutrition’ can be referred to as ‘malnutrition in women’; similarly, ‘diarrheal children’ – ‘children with diarrhea’

Response: Thank you for your valuable comments. We have revised the language style in the abstract section of the manuscript accordingly.

2. Figure 1 – why maternal job, family income, maternal/second hand smoking, micronutrient supplementation (both for mother and child), and birth weight, are not taken in to consideration.

Response: Thank you for your thoughtful observation. In the ABCD trial maternal job, family income, maternal/secondhand smoking, and micronutrient supplementation data were not collected. Birth weight data were collected retrospectively from the mother/caregiver, so there might be a chance of recall bias and only in 18% of cases the birthweight data was available. Because of these reasons we excluded the “birth weight” variable from our analysis.

3. For the children in 6-24 months category, mid-arm circumference would have been a good indicator. Why this was not included in the ABCD trial?

Response: thank you for your kind suggestion. We agree with your concern regarding the mid-upper arm circumference (MUAC) issue. Actually, in the ABCD trial, we enrolled patients from the 2-24 months’ age group having diarrhea and measured the MUAC on enrollment and in the subsequent two follow-ups. But the MUAC of below 6 months old children is not the true representation of child malnutrition. In our dataset, we have collected data from the children who were between the age group 2-6 months. That is why we exclude the MUAC data from our final analysis, which might be misleading for defining malnutrition among under 6 months old children.

Reviewer 2 Report

This study examined the association between maternal undernutrition and child undernutrition using CIAF. It showed the importance of maternal nutrition and maternal education to prevent child undernutrition. This study used data from ABCD trial.

The method is reasonable, but the presentation of the results needs to be improved. The unit is missing. Please add (%) to the unit on the horizontal axis. The authors also show the percentages in the table with two decimal places, but one decimal place is preferable.

Author Response

This study examined the association between maternal undernutrition and child undernutrition using CIAF. It showed the importance of maternal nutrition and maternal education to prevent child undernutrition. This study used data from ABCD trial.

The method is reasonable, but the presentation of the results needs to be improved. The unit is missing. Please add (%) to the unit on the horizontal axis. The authors also show the percentages in the table with two decimal places, but one decimal place is preferable.

Response: Thank you for your valuable comments. We have provided the units and (%) in the horizontal axis. We have replaced the table percentage in the table as one decimal point which is preferable as you mentioned.